# An Alternative Source of Biopesticides and Improvement in Their Formulation—Recent Advances

**DOI:** 10.3390/plants11223172

**Published:** 2022-11-20

**Authors:** Dragana Šunjka, Špela Mechora

**Affiliations:** 1Faculty of Agriculture, University of Novi Sad, Trg Dositeja Obradovića 8, 21000 Novi Sad, Serbia; 2Agency for Radwaste Management, Litostrojska 58A, 1000 Ljubljana, Slovenia

**Keywords:** biopesticides, organic waste, formulation, improvement

## Abstract

Plant protection in contemporary agriculture requires intensive pesticide application. Their use has enabled the increase in yields, simplifying cultivation systems and crop protection strategies, through successful control of harmful organisms. However, it has led to the accumulation of pesticides in agricultural products and the environment, contaminating the ecosystem and causing adverse health effects. Therefore, finding new possibilities for plant protection and effective control of pests without consequences for humans and the environment is imperative for agricultural production. The most important alternatives to the use of chemical plant protection products are biopesticides. However, in order to increase their application and availability, it is necessary to improve efficacy and stability through new active substances and improved formulations. This paper represents an overview of the recent knowledge in the field of biopesticides and discusses the possibilities of the use of some new active substances and the improvement of formulations.

## 1. Introduction

It is estimated that by 2050, agricultural production will have to increase by 70% in order to feed the growing population [1]. However, food production is a great challenge due to climate change, lack of fresh water, reduction of arable land, and particularly the presence of pests and diseases [2,3,4,5,6,7,8]. In order to achieve these goals, appropriate plant protection is required. At the same time, growing concerns about food safety, the trend of organic agricultural production, the presence of resistant pest populations, and the disruption of biodiversity as a result of intensive use of chemical pesticides are some of the main challenges of modern science. 

A particularly important issue in contemporary agriculture, from the aspect of plant protection, is the control of invasive species. Due to global travel and trade, pests appear earlier in the season and away from their native environments where they have not been introduced naturally [2]. In newly introduced habitats they can cause great damage to indigenous plant species and the environment. As a result of invasive expansion in the past, insects such as grape phylloxera (*Daktulosphaira vitifoliae* (Fitch)) [9] and the Colorado potato beetle (*Leptinotarsa decemlineata* (Say)) [10] were introduced from the United States into Europe, where they originally never appeared before. More recently, newly introduced species are *Drosophila suzukii*, native to East Asia [11], and the brown marmorated stink bug *Halyomorpha halys* (Stål) [12] and *Lycorma delicatula* (White), originating from northern China, which pose a serious threat to vines, apple, plum, and pear, but also to ornamental and forestry production [13]. Some of the examples of pathogens with a strong negative impact are the fungi *Phytophthora infestans* ([Mont] de Bary) [14] and *Puccinia graminis* f. sp. tritici (Pgt) [15]. Although a native species to Europe, the small bacteria, Flavescence Dorée phytoplasma [16], has in recent decades come to be considered an invasive and quarantined pest in European viticulture. The introduction of the American grapevine leafhopper (*Scaphoideus titanus* (Ball)), a vector whose life cycle is closely linked to the vine, has allowed rapid spread, and a negative impact on viticulture had been reported. The reduction of crop yield is also affected by weeds. Although weeds are a habitat for some beneficial organisms, a food source for pollinators, and contribute to reducing soil erosion, they are defined as “unwanted plants”, which makes weeds necessary to eliminate from agricultural ecosystems. One of the typical examples is the spread of allochthonous species *Ambrosia artemisiifolia* L., *Cuscuta campestris* Yunk., *Erigeron canadensis* L., and others [17]. In addition to insects, plant pathogens, and weeds, extensive damage to crops can also be caused by nematodes. 

Besides the direct impact on yield, the side effects of pests manifest after harvesting as well as a result of the presence of mycotoxins in food and feed, which can seriously endanger human and animal health [18]. The presence of pathogens, weed seeds, eggs, and larvae of insects, nematodes, etc., endanger agricultural products as well. Although these are just some examples, they confirm the significance and magnitude of the resulting damage. 

All the above-mentioned require intensive crop protection. However, modern agriculture almost completely relies on the use of chemical compounds, i.e., pesticides. Their use has enabled the increase in yields, simplifying cultivation systems and crop protection strategies through the successful control of harmful organisms. If pesticides would be totally forbidden, in only one year diseases, pests, and weeds would reduce world food production by 17–20% [19]. Nevertheless, their long-term use has led to a situation in which farmers have almost completely stopped using traditional methods of plant protection and replaced them with pesticides. 

However, the intensive and/or inappropriate use of chemicals causes ecosystem contamination and adverse health effects; pesticide residues in food endanger human health [20,21], along with their accumulation in the environment, affect non-target organisms in a negative way [22], and endanger the future of plant production by the development of pest resistance [23]. Moreover, the impossibility of applying synthetic pesticides during the ripening and harvesting period (especially in greenhouses) complicates plant protection even more. These are only some of the side effects of pesticide application. Despite this, the exporting of synthetic pesticides banned in the European Union (EU) to developing countries is still a big issue that needs to be addressed, especially given the fact that these chemicals pose serious and long-term risks to human health and the environment [24]. Therefore, there is a need to find new possibilities for plant protection and effective control of pests without consequences for humans and the environment, as imperative for agricultural production. The most important alternative to the use of chemical plant protection products are biopesticides, naturally originated compounds with pesticide activity. 

Biopesticides are also an important part of sustainable agriculture, which fulfills the United Nations 2030 agenda [25]. However, in order to increase their application and availability, it is necessary to improve efficacy and stability through new active substances and advanced formulations. 

This paper represents an overview of the recent knowledge in the field of biopesticides with the main objective being to highlight the possibilities of the use of some new active substances and formulations. 

## 2. Biopesticides—Advantages and Disadvantages 

The term biopesticides refers to the application of microbiological, biochemical, and macrobiological plant protection products for the control of pests, weeds, and disease-causing agents (Table 1). Microbiological pesticides contain selected genres of specific species or mixtures of different bacteria, fungi, viruses, or protozoa. These beneficial organisms produce toxins, vitamins, enzymes, and plant hormones that can act antagonistically on disease-causing, harmful insects, nematodes, and weeds. Additionally, beneficial microorganisms produce vitamins, enzymes, and plant hormones that can have an effect on plants’ immune systems by increasing their resistance. They are widely used and account for about 30% of the total production and sale of biopesticides. The specific mode of action of microorganism-based biopesticides relies on the competence for space and food, direct antagonism in relation to the growth of the target organism, and immunization of the host plant [26]. Microorganisms in plant protection products must have strong power to compete with the autochthonous microbial population and a high degree of ability to survive and adapt to the newly created conditions in which they should achieve their best efficiency. 

Biochemical pesticides are substances of natural origin that control harmful organisms through non-toxic mechanisms. They are produced by plants, animals, minerals, insects, etc. The most important biochemical pesticides are botanical pesticides, such as plant extracts and essential oils, i.e., plant derivatives. Plant extracts are chemicals or mixtures of chemicals obtained from higher plants. They usually contain various types of metabolites, including alkaloids, phenols, terpenoids, and secondary substances developed by plants in order to help them in the protection from the harmful insect. These products are characterized by a variety of compositions and modes of action, which have an influence on insects by repelling or exhibiting insecticidal effects. Plant extracts offer an unlimited resource of biodegradable, economical, and renewable alternative pest control measures. Essential oils are natural substances, a complex mixture of lipophilic, liquid, fragrant, and volatile components located in the secretory structures of aromatic plants, which are often responsible for the characteristic smell or taste. They exhibit physiological functions with hormonal action, keep coenzymes in a reduced form, and represent a source of energy. They also have an ecological function that is reflected in the reduction of respiration, creating a specific microclimate that protects plants from excessive transpiration, reflection, and refraction of light and participates in the plant–plant, plant–animal, and plant–insect interactions. These compounds are responsible for attracting insects, which is important for pollination. They can inhibit the germination of seeds of other and their own species, protect the plant from insect and animal attacks, and protect the plant from infection by microorganisms. Essential oils accumulate in all types of vegetative organs such as flowers, leaves, bark, tree, roots, rhizomes, fruits, and seeds. So far, thousands of compounds from the terpene class in essential oils have been identified [27].

Even macrobiological pesticides must not be left out. Macrobiological pesticides include zoophagous species such as insects, mites, and nematodes, which, depending on their way of life, are divided into predators, parasites, and parasitoids, also called natural enemies. Although natural enemies are already present in the environment, often for their successful activity in agroecosystems, it is necessary to introduce or apply them regularly when the need arises.

Can biopesticides suppress the use of chemical pesticides as the dominant strategy for the control of harmful pests in the future? Unrealistic expectations, problems with quality control, short shelf life, lack of awareness of their importance, and relatively high costs compared to the pesticide used in conventional agriculture are the main reasons for their insufficient application.

Despite this, the market of biopesticides has grown significantly in recent years due to increased awareness of their potential and focused attention on environmental and health risks associated with the use of synthetic pesticides. Biopesticides are only 4–5% of the global pesticide market, but it is estimated that this percentage could increase up to 20% in the near future [28], while according to some authors, by 2050 the importance of biopesticides in agricultural production will be equal to chemical pesticides [29]. It is difficult to achieve a complete replacement of chemical pesticides with biopesticides, considering the large number of harmful pests, invasive species, pesticide resistance, as well as climate change. However, it is reasonable to expect that biopesticides along with conventional pesticides will be included in the Integrated Plant Protection System (IPM). One example of Good Agricultural Practices is the integration of microbiological biopesticides into IPM with the aim of increasing the efficacy of natural agents, known as bio-intensive plant protection (BIPM) [30]. It has been proven that their efficacy increases in combination with other methods of integrated plant protection. To protect plants with the lowest risk, the use of biopesticides as an alternative to chemical pesticides is required. Plants, microorganisms, insects, and certain minerals are well-known sources of biopesticides, and according to the EPA, nowadays biopesticides include plant-incorporated protectants (PIPs) as well [31]. 

The most used biopesticides belong to the neem derivatives (azadirachtin) and *Bacillus thuringiensis* (Berliner), a Gram-positive bacteria with the capacity to produce a toxin that is completely harmless to humans, plants, and other animals but has great potential to be fatal to a wide range of insect pests, and the latest data indicate that 75% of biopesticides are based on *B. thuringiensis* [32]. Neem is a widespread botanical biopesticide [33]; however, due to its instability in the environment, its effectiveness is temporary. 

It is estimated that due to difficulties in isolating new organisms/compounds, the types of biopesticides will change. The intensive use of biopesticides in plant protection is limited by a number of risks—limited spectrum, short supply of products, interaction with non-target organisms, the virulence of strains, etc. Registration of bioinsecticides requires demanding protocols, causing a low rate of research in the field of biopesticides.

In the European Union, the development of non-chemical alternatives to synthetic pesticides has been recognized and defined by the law with the aim of achieving their sustainable use and reduction of side effects on human health and the environment (2009/128/EC) [34]. However, only a few countries recognize biopesticides as a separate group of pesticides, which has led to inappropriate methods for assessing their efficacy and safety [35].

The main steps in biopesticides development are the selection of biocontrol agents and formulation technology [36]. This requires a continuous and comprehensive study of the living organisms’ responses (beneficial and side effects), both on various agrochemicals and biotechnology products, the discovery of new bioactive compounds, improvements of formulations, and mandatory risk assessment, not only for humans and animals but also for the environment.

## 3. Sources of Biopesticides

Recently, renewable resources of biopesticides have been recognized as the potential to overcome resource limitations and environmental pollution. Within a circular economy, organic solid wastes are considered to be a useful resource for obtaining value-added products [37,38,39].

### 3.1. Agricultural and Forest Waste as a Source of Biopesticide Compounds

The use of agricultural and forest waste as a potential bio-resource for the production of pesticide compounds has been enabled by thermochemical processing (Table 2). From the aspect of sustainable use of waste and biomass, pyrolysis technology is being actively studied. This technology is not novel, since the use of pyrolysis products dates way back to the Middle Paleolithic times [40]. Depending on the process, pyrolysis can be slow or fast, resulting in liquid (bio-oil), solid (bio-char), and gaseous (syngas) products. However, there is insufficient scientific evidence for supporting claims regarding the efficacy and toxicological assessment of the used products. 

Bio-oil obtained by lignin pyrolysis at 450 and 550 °C shows insecticidal, fungicidal, and bactericidal activity [41]. Birchwood, pyrolyzed at 380 °C, in the form of birch tar oil mixed with Vaseline^®^ has a strong repellent effect against mollusks [42], which makes it an effective, cheap, and simple method for controlling mollusks. Furthermore, the pesticide activity of liquids obtained from slow pyrolysis (pyroligneous acids) and hydrothermal carbonization (HTC) products has been stated [40,41,43,44]. However, there are numerous challenges in the development of new biocontrol technologies and barriers to their commercialization. The characterization of pyrolysis liquid showed that polycyclic aromatic hydrocarbons (PAHs) mainly contribute to pesticide activity [41,43]. They are major environmental pollutants with toxic, mutagenic, and carcinogenic effects on various organisms; thus, their release into the environment is highly restricted. Nevertheless, by reducing the temperature for the separation of liquids during pyrolysis below 300 °C, the appearance of PAHs and tar is reduced, which can be one of the possible directions for development and potential application [45,46]. Studying the pesticidal activity of various liquids obtained by slow pyrolysis of pine bark, pine forest waste, wheat straw, willow, and hydrothermally carbonized (HTC) willow, at temperatures of 280 °C and 260 °C, showed high efficiency of slow pyrolysis liquid from willow on *Arianta arbustorum* (L.) (repellent action), *Brassica rapa* (L.) (herbicidal action), and *Rhopalosiphum padi* (L.) (insecticidal action) [47]. A slightly lower efficacy was found in the liquid from wheat, bark, and forest residues. The pesticide activity of PAH-free liquids from slow pyrolysis originates from simple organic volatile compounds and numerous phenolic compounds. The main reasons for the higher pesticide activity of willow-derived pyrolysis liquid are high levels of acetic and other carboxylic acids, as well as the presence of dozens of different phenolic compounds. Thus, slow pyrolysis liquids represent a potential source of biopesticides, but there is a necessity for the optimization of technology production. 

As already stated, an important source of bioactive compounds are plants’ secondary metabolites. The biological activities of phenolic and polyphenolic compounds in olive mill wastewater have already been confirmed for the control of several insect pests, mostly due to the presence of polyphenol oleuropein [48,49,50,51,52]. These compounds have potent bioactive properties as insecticides and growth regulators and can be successfully used for the control of the Mediterranean fruit fly *Ceratitis capitata* (Wiedemann) [53], a highly invasive species. 

A valuable agricultural by-product is grape pomace. In wine production, at least 20% of the fruit weight is discarded as this substance [54]. Grape pomace is used for the production of spirits and liquors, as fertilizer or animal feed, and even as composting material [55]. However, the presence of bioactive compounds, such as stilbenes resveratrol, in pomace extracts, indicates that grape pomace is a valuable source of biopesticides [56]. Due to its ability to inhibit the growth of bacteria, fungi, and viruses, resveratrol has been intensively studied [57,58]. Recently, resveratrol was proven to be effective in the control of *Botrytis cinerea* (Pers.), suppressing mycelial growth and conidia germination [59,60].

### 3.2. Solid Waste as a Medium for Microbial-Based Pesticides

Microbial-based biopesticides require a rich nutrient commercial medium for developing microbial species. In the process of production, this is a cost-consuming step. Production of microbial biopesticides using nonhazardous solid biodegradable waste, instead of commercial media, can contribute to solving the problem of increased waste production [61]. As an alternative to conventional media, kitchen waste could be used as a substrate for bio-pesticide production. Results showed that kitchen waste in a combination with wheat bran, soybean cake power, grain hulls, and mixed ions is suitable for the growth of *B. thuringiensis* [62,63], the most used biocontrol agent worldwide.

Biodegradable wastes from agriculture, industry, and households are being studied in order to obtain biopesticides through solid-state fermentation. At the laboratory level, it was confirmed that biowaste (source-selected organic fraction of municipal solid waste) is a substrate suitable for *B. thuringiensis* growth under non-sterile conditions [39,64]. Some authors provided results of using chicken feathers as a medium for the production of entomopathogenic bacteria (*Bacillus thuringiensis* serovar *israelensis* (Bti) and *Bacillus sphaericus* (Meyer and Neide) (Bs)) [65].

Furthermore, fungal-based biocontrol agents, such as the genus *Beauveria* and *Trichoderma*, are also highly promising alternatives to traditional chemical pesticides. Optimization of the fermentation process would allow the use of agroindustrial waste, rice husk, as a source of fungal production [66].

As a potential source of biopesticides and biofertilizers, some authors state the use of microalgae due to the production of biologically active compounds with antimicrobial activity [67,68]. An added value represents the possibility of using wastewater as a growing medium given that they require nitrogen, phosphorus, carbon, and ammonium [69].

The release of huge amounts of bioorganic waste from the food industry, poultries, and fisheries [65] is an opportunity for biopesticide production, either as a medium or as a source of bioactive compounds.

## 4. Advances in Formulations

Biopesticides are considered a powerful tool for developing a more rational strategy for the use of pesticides, which should lead to the improvement of the balance between efficacy, production costs, and application [70]. The history of biopesticides, classification, and mode of action are well known and extensively studied [71,72,73,74]. However, in order to ensure its successful application, the efficacy and achievement of higher formulation stability still have to be improved.

The formulation of biopesticides needs to ensure the stability of the organism, i.e., the compounds included in the preparation composition during production, distribution, storage, handling, and application of the preparation. Moreover, it is necessary to protect the biological agent/compound from the external environment influence, as well as enable an increase of the organism’s activity during its reproduction, contact, or interaction with the target organisms. All of the above are achieved by adding proper non-pesticide compounds [31].

The formulation of such products poses a challenge, considering that such formulations must meet a number of objectives such as satisfactory efficiency, environmental acceptability, constancy after application, and uniform distribution throughout the treated object.

The fact that inert ingredients enhance biopesticide activity has opened new opportunities for further development in this field. A suitable formulation can improve the stability of the plant protection product and increase or expand the activity with a reduction of the influence of external factors [75].

For example, essential oils, regardless of origin, can be phytotoxic if applied to plants in high concentrations. Compared to conventional insecticides, insecticides based on essential oils are less effective and therefore must be applied at relatively high concentrations, often in the range of 0.5–1.5%. However, the procedure for bioinsecticide registration must include an assessment of the phytotoxic effects on the crops with special care. Little or no residues are expected in the agricultural product after the application of essential oil-based pesticides, and the potential impact on the organoleptic quality of the treated products has not been studied in detail to date [28].

A relatively new area that finds its use in agriculture and the food industry is nanotechnology [76]. This technology has enabled the development of controlled-release formulations, such as nanoemulsions, nanosuspensions, nanocapsule suspensions, etc. [77].

Besides the improvement of the bioactivity of synthetic pesticides, the use of nano-technologies in plant protection enables the limitations of biopesticides to be overcome [78]. Due to their unique properties, nanomaterials are considered suitable carriers for stabilizing fertilizers and pesticides, as well as for facilitating the controlled transfer of nutrients and increasing plant protection. Thus, stability, persistence in the environment, and toxicity to target organisms would be improved, while the side effects and phytotoxicity would be reduced [79]. At the same time, nanotechnology-based pesticides, especially biosynthesized and bioinspired materials, make a huge contribution to sustainable development [80] due to their controlled release of the active ingredients [81], ensuring their efficacy in long-term use with the possibility of resolving the issue of accumulation of pesticide residues [82].

Despite the promising and well-known pesticidal activity of botanical pesticides, only a few essential oil-based biopesticides are available [28,82,83]. The main disadvantage of the commercialization of these biopesticides as plant protection products is their high volatility, which limits application with a relatively short activity in the field, requiring repeated applications [84]. The possible ways to overcome these limitations can be achieved with nanotechnologies; the essential oil needs to be enclosed within nanoparticles or nanoemulsions, which allow their stability and dispersibility in water [85]. Nanoencapsulation is based on encapsulating EOs in materials in the order of nanometers.

Along with botanical pesticides, microbial pesticides could express better pesticide activity for some pests in the field; however, they show activity against only one type of pest. This is one of the biggest disadvantages of microbial pesticides [86]. Environmental factors such as desiccation, heat, light, and UV reduce the activity of microbial pesticides, causing continuous crop destruction [87]. Among the newer technologies for the production of biopesticides based on microorganisms is bioencapsulation technology. Encapsulation of microorganisms in microcapsules has significant survival benefits, while also ensuring the controlled release of these bacteria throughout the growing season [75,88,89,90]. Encapsulation involves the active ingredient being enclosed within the polymer. The size of the capsule, which provides controlled release of the active ingredient after plant protection product application, varies from 2–50 μm, or 1–2 μm [91,92]. For encapsulation of micro-organisms, various materials are used, including natural and synthetic polymers such as agar and agarose, starch, corn syrup, polyacrylamide, and polyurethane from artificial materials [93,94]. The capsule does not erode during the release process. The pores close again when the capsule is exposed to osmotic stress/dehydration. Capsules can be stored at room temperature, and storage time can be significantly extended by adding nutrients to the capsule.

However, sustainable nanotechnology requires a science-based environmental risk assessment [95]. The production of nanoparticles needs hazardous materials and advanced, modern equipment and exhibit side effects on the environment. Based on several toxicological studies, concerns have emerged about the safety of nanomaterials and the side effects on the environment [96]. Micro/nanoemulsions are stable systems with the possibility of spreading on the plant surface, which increases bioactivity, but at the same time, this increases phytotoxicity [97]. Therefore, over the last decade, research has shifted towards environmentally friendly and sustainable, more economical, “green” synthesis with the aim of supporting the growing use of nanoparticles in various industries. Green synthesis, as part of bioinspired protocols, provides reliable and sustainable methods for nanoparticle bio-synthesis using a wide range of microorganisms rather than current synthetic processes [98].

A good example is a formulation of nanoemulsions based on commercial essential oils (EO) using polyoxyethylene sorbitan monooleate and water [82]. By adding distilled water and agarose to the EO-based nano-formulations, followed by the addition of sodium polyacrylate, EO-based nanogels can be formulated. The results proved optimal physical characteristics of essential oil-based nanoemulsions and insecticidal activity against the test insect *Tribolium confusum* (Jacquelin du Val).

The application of biosynthesis as a tool for the design and development of many nanomaterials has become a sustainable and eco-friendly method [80]. Myco-nanotechnology, based on the development of nanoparticles employing fungi [94], and many other biological systems, such as bacteria, yeast, actinomycetes, and plant parts, have shown promising suitability for nanomaterial biosynthesis and represent a greener alternative to chemically synthesized nanoparticles [99].

One of the main limiting factors of biopesticide application is UV sensitivity. In the formulation of biopesticides based on baculovirus as an active ingredient, this limitation was overcome using titanium dioxide as a UV absorbent. Sensitive viral DNA was protected by the ENTOSTAT wax capsule, which dissolved in the alkaline intestines of insects to release the virus. Moreover, this prolongs the efficacy and stability of biopesticides without side effects on crops [75].

Improvements in the formulation of biopesticides based on *Bacillus thuringiensis* using starch-producing industry wastewater could be successfully carried out with the addition of a soybean medium [100].

## 5. Perspectives

In order to make the agricultural system safer and more sustainable, the European Commission announced plans to reduce the use and risk of chemical and more hazardous pesticides by 50% by 2030 [101]. In order to reduce the use of pesticides, in addition to numerous preventive measures, it is necessary to introduce biopesticides in plant protection. Currently, 60 biopesticide active substances are available on the EU market, while the number in the United States market is over 200 [102]. Such a small number of registered biopesticide plant protection products is a consequence of limited resources of bioactive agents, demanding registration procedures, etc.

In order to fulfill the goal, it is necessary to intensify the development of biopesticides and overcome their main shortcomings by choosing proper active agents and improving formulations. This primarily refers to increasing the specificity and longevity of the plant protection product, reducing the effective dose, and improving the speed of activity. A shorter shelf-life not only reduces the efficacy of biopesticides but also reduces their competitiveness with chemicals.

Although biopesticides include a wide range of living organisms, products of their metabolism, and compounds of plant origin, they have completely different characteristics and activities in specific ecosystems. This requires extensive scientific research in order to create the conditions for the transition to the phase of commercialization and wider use of such preparations [103]. At the same time, the main limitation of biopesticides commercialization is the strict regulations concerning their placement on the market. Long-term demanding procedures with insufficiently defined or maladapted principles are the main obstacles to the more significant development of the biological products industry for plant protection. In addition, not recognizing the difference between living organisms and bioactive compounds represents one of the biggest problems when registering biopesticides [104].

In view of the above stated reasons, the imperative is finding an optimal solution that would simplify the procedures and enable the development of new biopesticides, followed by the release of plant protection products on the market with a competitive price acceptable to agricultural producers [102], and thus an increase in their application [77].

Until the categories of “basic substances” and “low-risk substances” were introduced, biopesticides did not have a regulatory category [105]. Today, for basic substances not primarily intended for plant protection products, but which may provide crop protection possibilities, an application authorization is granted for the whole EU for an indefinite period. Specific to low-risk substances is that their application can be partially approved based on data from the literature and scientifically justified opinions. In this case, it is assessed whether microbiological and semiochemical products (e.g., pheromones) comply with low-risk criteria, in an assessment that only lasts for 120 days, while the approval may be valid for 15 years instead of 10 years. For many biopesticides, the components of the formulation are inert or not toxicologically significant, and the risk assessment can be based only on the active substance and on scientific evidence.

Limited sources of biopesticides are one of the most significant challenges. Finding new sources of bioactive compounds with an emphasis on renewable sources is imperative for modern plant protection. However, most of the research conducted in the field of biocontrol has provided only empirical results at the laboratory level. Many of these research studies will never be continued in real environmental conditions, primarily because the commercialization of biopesticide products requires a continuous and easily accessible source of bioactive components and a suitable formulation.

Furthermore, the formulation of biopesticides represents another crucial challenge. Production of biopesticides is more expensive, and their use is often more technically complex than the chemical ones. Thus, it is the main barrier to biopesticide placement on the market. However, research on their production, formulation, and application could greatly help in the commercialization of biopesticides. With the development and improvement of biopesticide formulations, the balance between efficiency, production costs, and application will be improved [70], which will lead to intensifying their application in the future.

## 6. Conclusions

The finding of alternative sources of compounds with pesticide activity is the most challenging issue in the field of plant protection. To overcome resource limitations and environmental pollution, renewable resources play an important role. Aside from plants as a source of bioactive compounds, the use of different types of waste as a medium for microbial growth is a significant source of biopesticides. This also could contribute to solving the problem of increased waste production. Although significant progress has been made in the development of the formulations and methods of application, further research is necessary regarding the use of biopesticides in plant protection. This is especially aimed at the need for a legal framework that would regulate nanomaterials placed on the food market. Future research will aim at the improvement of techniques and multidisciplinary research that will provide good, safe, effective, and inexpensive plant protection products.

## Figures and Tables

**Table 1 plants-11-03172-t001:** Biopesticides.

Microbiological Pesticides	Biochemical Pesticides	Macrobiological Pesticides
Bacteria	Plants	Insects
Fungi	Animals	Mites
Virus	Minerals	Nematodes
Protozoa	Insects	

**Table 2 plants-11-03172-t002:** Organic waste as a source of biopesticide compounds.

Source of Biopesticide	Authors
Bio-oil (fast pyrolysis of biomass)	[41]
Birch tar oil (fast pyrolysis of biomass)	[42]
Pyroligneous acids (slow pyrolysis of pine bark, pine forest waste, wheat straw)	[40,41,43,44]
Olive mill wastewater	[48,49,50,51,52,53]
Grape pomace	[54,56,57,58,59,60]

## Data Availability

Not applicable.

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
