# Peer review of "An Alternative Source of Biopesticides and Improvement in Their Formulation—Recent Advances"

_plants, 2022, doi:10.3390/plants11223172_

Round 1

Reviewer 1 Report

This is an interesting review paper for the field of non-chemical methods in plant protection. In the paper the authors present an overview of alternative sources for the production of biopesticides. The paper is written according to the guidelines for writting a review scientific paper and contains over 100 references. The structure of the paper is suitable. I suggest to publish a paper after minor revision, since I have found the following issues that need to be corrected in the paper:

p. 1, lines 24-26: More references should be mentioned, for example:  ADLER et al., 2022. Changes in the distribution and pest risk of stored product insects in Europe due to global warming : Need for pan-European pest monitoring and improved food-safety. Journal of Stored Products Research,  97, art. 101977, 9 p.  

p. 1, line 36: replace "phylloxera" with "grape phylloxera". It is important to use the official common names of the organisms! This should be taken into account throughout the paper! 

p. 1, line 36: when an organism is first mentioned in the text, it should be mentioned with full Latin name, i.e. Daktulosphaira vitifoliae (Fitch). This should be taken into account throughout the paper! 

p. 2, line 46: replace "American cricket" with "American grapevine leafhoper". Also the suitability of common names of all other organisms mentioned should be taken into account!

p. 4, line 157: replace "Bacilus" with "Bacillus"

p. 6, line 250: "serovar" should not be written in italic

p. 6, lines 252-253: "Beauveria" and "Trichoderma" should be written in italic

Author Response

Dear Reviewer #1,

Thank you very much for your careful review and for the constructive comments that significantly improved our paper ''An Alternative Source of Biopesticides and Improvement in Their Formulation - Recent Advances''.

We have studied them carefully and made a correction which we hope meets with your approval.

All corrections are made in the revised manuscript, submitted with these comments.

In the following text, please find a point-by-point response to the comments:

  1. 1, lines 24-26: More references should be mentioned, for example: ADLER et al., 2022. Changes in the distribution and pest risk of stored product insects in Europe due to global warming : Need for pan-European pest monitoring and improved food-safety. Journal of Stored Products Research, 97, art. 101977, 9 p. 

  • In order to better explain the impact of different factors on food production, in the above-mentioned paragraph, we have added references under numbers 3, 4, 5, 6, 7, and 8.

  1. 1, line 36: replace "phylloxera" with "grape phylloxera". It is important to use the official common names of the organisms! This should be taken into account throughout the paper!
  • Oversight made within common names in this manuscript is completely corrected.
  1. 1, line 36: when an organism is first mentioned in the text, it should be mentioned with full Latin name, i.e. Daktulosphaira vitifoliae (Fitch). This should be taken into account throughout the paper!
  • According to your instructions, the Latin names of species mentioned in the manuscript are corrected.
  1. 2, line 46: replace "American cricket" with "American grapevine leafhoper". Also the suitability of common names of all other organisms mentioned should be taken into account!
  • Common names in this manuscript are corrected in accordance with your suggestions.
  1. 4, line 157: replace "Bacilus" with "Bacillus"
  • We apologize for the spelling mistake. It is corrected.
  1. 6, line 250: "serovar" should not be written in italic
  • It is corrected.
  1. 6, lines 252-253: "Beauveria" and "Trichoderma" should be written in italic
  • It was written in italic.

Reviewer 2 Report

I would add some figures in the paper or table to semplify the reading. E.G. a table of pyrolisis temperature and system on different experiment, figures of the different possible pesticedes.

Moreover I wpuld move lines 133-179 before perspective and I creates a new paragraph about fure of pesticides

Author Response

Dear Reviewer #2, 

Thank you very much for your careful review and for the constructive comments that significantly improved our paper ''An Alternative Source of Biopesticides and Improvement in Their Formulation - Recent Advances''.

We have studied them carefully and made a correction which we hope meets with your approval.

In the following text, please find a point-by-point response to the comments:

I would add some figures in the paper or table to semplify the reading. E.G. a table of pyrolisis temperature and system on different experiment, figures of the different possible pesticedes.

  • Due to your suggestions, we have added two tables. In the first table, we provide the main types of biopesticides used nowadays, while in the second one in the tabular form we showed the possibilities of using organic wasters as a new source of biopesticides.
  • All corrections are made in the revised manuscript, submitted with these comments.

Moreover I wpuld move lines 133-179 before perspective and I creates a new paragraph about fure of pesticides

  • In paragraph 2. Biopesticides, including text in lines 133-179, we explained the recent status of biopesticides, their advantage, and disadvantage. In the last part of this paragraph, we stated possibilities to overcome these limitations such as new sources and improved formulations. This part is logically followed by paragraph 3. Source of biopesticides.
  • Moving text in lines 133-179, the structure of the paper will be changed.
  • Furthermore, there is already a number of papers dealing with pesticide harmfulness.